# ASSESSING GENERALIZATION IN TD METHODS FOR DEEP REINFORCEMENT LEARNING

## ABSTRACT

Current Deep Reinforcement Learning (DRL) methods can exhibit both data inefficiency and brittleness, which seem to indicate that they *generalize* poorly. In this work, we experimentally analyze this issue through the lens of *memorization*, and show that it can be observed directly during training. More precisely, we find that Deep Neural Networks (DNNs) trained with supervised tasks on trajectories capture temporal structure well, but DNNs trained with TD(0) methods struggle to do so, while using TD($\lambda$) targets leads to better generalization.

## 1 INTRODUCTION

Deep neural networks (DNNs) trained on supervised learning tasks using i.i.d. data have shown the capacity to learn quickly even from a small amount of samples (Hardt et al., 2016). Intuitively, this is due to each sample also providing information about the estimate corresponding to other samples; research suggests that DNNs first extract structures that are informative of the modes of the data (even if later on they can also memorize see Zhang et al. (2016); Arpit et al. (2017)), and that they can transfer well (Yosinski et al., 2014; Li et al., 2015), even from relatively few samples. In contrast, in Deep Reinforcement Learning (DRL), the number of samples required for an agent to learn successfully is often very high; many modern algorithms struggle to perform well until they acquire tens of millions of samples (Mirowski et al., 2016; Vinyals et al., 2017; Hessel et al., 2018), and some even diverge to bad solutions (Anschel et al., 2017). While there are many facets to sample complexity and brittleness, we posit that a contributing factor is a lack of what we call **gradient update generalization**, i.e., *whether performing updates at one state provides useful information about the value/policy at other states*.

**Generalization in RL** is of two types: (a) generalization to unseen states–will an agent trained on a single MDP pick the optimal action for a state it has never seen before? (b) generalization to unseen tasks–will an agent trained on a distribution of MDPs know how to act in an MDP it has never seen before? Both of these facets are actively studied. For example, Farebrother et al. (2018) expose some generalization failures on the Atari domain (Bellemare et al., 2013) and study the impact of regularization, Zhang et al. (2018) study the generalization capabilities of DRL agents on randomized mazes, Packer et al. (2018) study the extrapolation capabilities of DRL agents trained on a distribution of environment parameters (e.g. pole mass in CartPole) outside of the training distribution, Cobbe et al. (2018) find that even on procedurally generated environments, DRL agents can easily overfit on their training set unless regularized, Oh et al. (2017) study the embedding regularizations necessary for agents to generalize to new instruction sequences on navigation tasks.

In this study, we are not interested in measuring state generalization (i.e. predictions for unseen states), nor task generalization (i.e. in terms of the quality of the behaviour), but rather generalization *within* the process of stochastic gradient learning. In other words, since any kind of generalization must arise through the accumulation of parameter updates, it seems useful to measure whether these parameter updates are themselves *general*. To this end, we propose the measure of **gradient update generalization**, best understood as a side-effect of neural networks sharing parameters over their entire input space. That is, updating parameters after seeing one state will change the prediction for virtually all other states; we are interested in measuring that change.

**TD methods** are a broad class of RL algorithms that form a target for an update by utilizing the current estimate of the value function. They include TD(0) and TD($\lambda$) methods for estimating the value of a fixed policy, as well as Sarsa and Q-learning algorithms for control. TD methods have

achieved success in some challenging tasks (Tesauro, 1995; Mnih et al., 2013; Hessel et al., 2018), but they are also known to have problems when coupled with function approximation (Sutton, 1995; Baird, 1995; Tsitsiklis & Van Roy, 1997; Chung et al., 2018). Previous studies explicitly addressed problems such as leakage propagation in TD (Penedones et al., 2018), while others aimed to provide sampling improvements (Schaul et al., 2015; Andrychowicz et al., 2017; Fu et al., 2019), explicit temporal regularization (Thodoroff et al., 2018), or auxiliary tasks which push the agent to learn more about the temporal structure in the data (Jaderberg et al., 2016).

To our knowledge, no study to date has focused on the dynamics of the generalization process itself, within TD-based DRL methods[1] such as deep Q-Learning (Riedmiller, 2005; Mnih et al., 2013), Sarsa (Rummery & Niranjan, 1994), and TD($\lambda$) (Sutton, 1988; Schulman et al., 2015). For this study, we introduce the aforementioned measure of **gradient update generalization**, which enables us to differentiate the learning behaviours of different methods. Overall, we find that:

1. when doing a TD(0) update for a single state, parameters change in such a way that the value prediction of other states is generally not affected, surprisingly even for states that are close either temporally or in an annotated "ground truth" state space;
2. DNNs trained with TD(0), in contrast with DNNs trained on a memorization task or using a supervised objective, do not entirely memorize their state space, yet also do not generalize in the way we would expect;
3. both the choice of optimizer and the nature of the objective impact the generalization behaviours of models; in particular, when increasing the $\lambda$ parameter in TD($\lambda$), DNNs appear to capture more temporal structure.

## 2 TECHNICAL BACKGROUND

A Markov Decision Process (MDP) (Bellman, 1957; Sutton & Barto, 2018) $\mathcal{M} = \langle S, A, R, P, \gamma \rangle$ consists of a state space $S$, an action space $A$, a reward function $R : S \to \mathbb{R}$ and a transition probability distribution $P(s'|s, a)$. RL agents aim to optimize the expectation of the long-term return:

$$G(S_t) = \sum_{k=t}^{\infty} \gamma^{k-t} R(S_k). \tag{1}$$

where $\gamma \in [0, 1)$ is called the discount factor. Policies $\pi(a|s)$ map states to action distributions. Value functions $V^\pi$ and $Q^\pi$ map states/states-action pairs to expected returns, and can be expressed recursively:

$$V^\pi(S_t) = \mathbb{E}_\pi[G(S_t)] = \mathbb{E}_\pi[R(S_t) + \gamma V(S_{t+1})|A_t \sim \pi(S_t), S_{t+1} \sim P(S_t, A_t)] \tag{2}$$

$$Q^\pi(S_t, A_t) = \mathbb{E}_\pi[R(S_t) + \gamma \sum_a \pi(a|S_{t+1})Q(S_{t+1}, a)|S_{t+1} \sim P(S_t, A_t)] \tag{3}$$

While $V^\pi$ could also be learned via regression to observed values of $G$, these recursive equations give rise to the *Temporal Difference (TD)* update rules for policy evaluation, relying on current estimates of $V$ to *bootstrap*, e.g.:

$$V(S_t) \leftarrow V(S_t) - \alpha(V(S_t) - (R(S_t) + \gamma V(S_{t+1}))), \tag{4}$$

where $\alpha \in [0, 1)$ is the step-size. Bootstrapping leads also to algorithms such as **Q-Learning** (Watkins & Dayan, 1992) and fitted-Q (Ernst et al., 2005; Riedmiller, 2005):

$$\mathcal{L}_{QL}(S_t, A_t, R_t, S_{t+1}) = [Q_\theta(S_t, A_t) - (R_t + \gamma \max_a Q_\theta(S_{t+1}, a))]^2, \tag{5}$$

**Sarsa** (Rummery & Niranjan, 1994):

$$\mathcal{L}_{Sarsa}(S_t, A_t, R_t, S_{t+1}, A_{t+1}) = [Q_\theta(S_t, A_t) - (R_t + \gamma Q_\theta(S_{t+1}, A_{t+1}))]^2 \text{ with } A_t \sim \pi(S_t) \tag{6}$$

---

[1]In contrast, policy-gradient algorithms such as PPO (Schulman et al., 2017) A3C (Mnih et al., 2016) and SAC (Haarnoja et al., 2018) are capable of learning good policies without necessarily having learned a good value function, and although interesting results have emerged to understand learning behaviours in policy-gradient methods (Ilyas et al., 2018), these methods build upon TD and analyzing them would add undesired confounders.

and **TD($\lambda$)**, which trades off between the unbiased target $G(S_t)$ and the biased TD(0) target (biased due to relying on the estimated $V(S_{t+1})$), using a weighted averaging of future targets called a $\lambda$-return (Sutton, 1988; Munos et al., 2016):

$$G^\lambda(S_t) = (1-\lambda)\sum_{n=1}^{\infty}\lambda^{n-1}\left[\gamma^n V(S_{t+n}) + \sum_{j=0}^{n-1}\gamma^j R(S_{t+j})\right] \tag{7}$$

$$\mathcal{L}_{TD(\lambda)}(S_t) = (V_\theta(S_t) - G^\lambda(S_t))^2 \tag{8}$$

(note that the return depends implicitly on the trajectory followed from $S_t$). When $\lambda = 0$, the loss is simply $(V_\theta(S_t) - (R_t + \gamma V_\theta(S_{t+1})))^2$, leading to the algorithm called TD(0) (Sutton, 1988).

## 3 UPDATE GENERALIZATION IN DEEP RL

We will now define the measure we propose in order to quantify the speed at which generalization to unseen states occurs, and to characterize the structure under which this generalization occurs. We define *gradient update generalization* as the expected improvement in the loss function $\mathcal{L} : \Theta \times \mathcal{X} \to \mathbb{R}$ after updating parameters $\theta \in \Theta$, on sample $X_U \in \mathcal{X}$, using update function $U_\mathcal{L} : \Theta \times \mathcal{X} \to \Theta$ (e.g. SGD or a semi-gradient methods like TD(0)):

$$Y_\mathcal{L}(X_U; \theta, U) = \mathbb{E}_\mathcal{X}[\mathcal{L}(\mathcal{X}; \theta) - \mathcal{L}(\mathcal{X}; U_\mathcal{L}(\theta, X_U))]. \tag{9}$$

If generalization from the samples in $X_U$ to $\mathcal{X}$ is good, this measure of gain should be large, and intuitively fewer other samples should be needed to achieve a desired level of performance. On the other hand, if on average the loss only decreases for the samples $X_U$ used in training, then more data in $\mathcal{X} - X_U$ will have to be visited before the model can learn. Hence, this measure is related to both sample complexity and the speed of learning (see Fig. 15 for empirical confirmation of this phenomenon).

As computing the exact expectation is usually intractable, we empirically measure gains on different subsets $X \subset \mathcal{X}$. In particular, when $X$ is chosen to be a slice around $X_U$ in the replay buffer, we write $Y^{\text{near}}$. We also subscript $Y$ with the corresponding loss' subscript, e.g. for (5), $\mathcal{L}_{QL}$, we write $Y_{QL}$. In this study, we are interested in TD-based methods that rely heavily on bootstrapping, Q-Learning, Sarsa, and TD($\lambda$), and measure $Y$ using their respective losses, (5), (6), and (8).

**Structure in DNNs** A common intuition in deep learning (Zhang et al., 2016; Arpit et al., 2017; Zhang et al., 2018) is that DNNs first learn about the *structure* of their data, meaning the underlying (usually linear) factors of variation of the data being mapped into the hidden units' space via parameter sharing. These factors of variation are usually conceptualized as a low-dimensional space where each dimension *explains* part of the data (Bengio et al., 2013). It is commonly assumed that a model which generalizes well will naturally capture these factors in the configuration of its parameters, in which case the gradient of the prediction w.r.t. all examples sharing the same latent factors of variation will be very close; updating with only one sample will change the prediction for all the related examples. Hence, a DNN which captures structure correctly should show high gradient update generalization.

**Temporal structure in RL** Data used in RL algorithms usually exhibits two additional types of structure: coherence of the inputs in a trajectory over time (e.g. pixel values in adjacent frames are often similar), and smoothness of the value function in time (in the sparse-reward case with $\gamma$ close to 1, $V(S_t) \approx \gamma V(S_{t+1})$, which is smooth in time, aside from rare discontinuities upon seeing rewards). Since RL data consists of trajectories which often have strong temporal structure of both types, we hypothesize that the gain $Y^{\text{near}}$ of temporally correlated examples should increase closer in time to the sample used in the update.

**Parameter sharing** Another indirect measure of update generalization related to parameter sharing is the **difference since last visit**, which we denote as $\Delta$. At each update iteration $k$, we compute the difference between the value $V_{\theta_k}(s)$ or $Q_{\theta_k}(s, a)$ predicted from the current parameters, $\theta_k$, and $V_{\theta_{last(s)}}(s)$ or $Q_{\theta_{last(s)}}(s, a)$ , i.e. the prediction made the last time state $s$ was used for a gradient update.[2] To illustrate, if $V_\theta$ was a lookup table, $\Delta$ would always be 0, while for a DNN, when states

---

[2] In practice, we simply cache the value prediction for all states in a replay buffer (as states in a continuous state space are unlikely to be encountered many times), and update the cache after a minibatch update (for those states only).

are aliased together, $\Delta$ should accurately reflect the effect of parameter sharing after performing *sequences* of updates (in contrast, (9) uses only a single update).

### 3.1 EXPERIMENTAL SETUP

We will now perform a series of experiments aimed at assessing the amount of generalization of various bootstrapping algorithms, compared to supervised learning, in combination with DNNs.

First, we test whether DNNs have a large gradient update generalization gain when trained under ideal conditions (data generated by expert policies and labelled with correct values, which can be used in supervised learning). Then, we test the policy evaluation case (using the same input data, but bootstrapped targets instead of supervised learning). We then test the usual control case, when no expert trajectories are available. Finally, we measure the effect of TD($\lambda$) on generalization gain in policy evaluation, as well as test Q-Learning's robustness to withheld data.

We perform our experiments on the Atari environment (Bellemare et al., 2013), with the stochastic setup recommended by Machado et al. (2018). We use a standard DQN architecture (Mnih et al., 2013). In order to generate expert trajectories, we use rollouts from a policy trained with Rainbow (Hessel et al., 2018); we denote $\mathcal{D}^*$ a dataset of transitions obtained with this agent, and $\theta^*$ the parameters after training that agent. For control experiments, we use Mnih et al. (2013)'s Q-Learning setup. When measuring $Y^{near}$ we choose the nearest 60 examples in time to a given state-action pair (30 previous and 30 following on the same trajectory).

### 3.2 ASSESSING TEMPORAL STRUCTURE WITH SUPERVISED LEARNING

In this experiment, we will assess if temporal structure, as described above, exists and can be captured by our architecture. To do so, we train DNNs starting from random parameters but with "ideal" targets coming from the expert parameters $\theta^*$ and expert trajectories $\mathcal{D}^*$; this removes all non-stationarity from the learning. We train $Q_\theta$ with 3 different objectives:

$$\text{MC:} \qquad \mathcal{L}_{MC}(s, a; \theta) = (Q_\theta(s, a) - G^{(\mathcal{D}^*)}(s))^2 \qquad (10)$$

$$\text{Reg:} \qquad \mathcal{L}_{reg}(s, a; \theta) = (Q_\theta(s, a) - Q_{\theta^*}(s, a))^2 \qquad (11)$$

$$\text{TD}^*: \qquad \mathcal{L}_{TD^*}(s, a, r, s; \theta) = (Q_\theta(s, a) - (r + \gamma \max_{a'} Q_{\theta^*}(s', a')))^2 \qquad (12)$$

where by $G^{(\mathcal{D}^*)}(s)$ we denote the Monte-Carlo return within the dataset $\mathcal{D}^*$, as in (1). Note that since $\mathcal{L}_{TD^*}$ "bootstraps" to $\theta^*$, this should be roughly equivalent to $\mathcal{L}_{reg}$, the latter being plain supervised learning (or some sort of distillation, *à-la* Hinton et al. (2012)).

Results are visualized in Fig. 1 for experiments ran on MsPacman, Asterix, and Seaquest for 10 runs each. Results are averaged over these three environments (they have similar magnitudes and variance). Learning rates are kept constant, they affect the magnitude but not the shape of these curves.

We draw two conclusions from these results. First, as seen in Fig. 1a & 1b, all curves tend to have large gains around $x = 0$ (the sample used in the update), especially from indices -10 to 10, showing that **there is some amount of temporal structure** captured by both objectives. Since $Q_{\theta^*}$ is a good approximation, we expect that $Q_{\theta^*}(s, a) \approx (r + \gamma \max_{a'} Q_{\theta^*}(s', a'))$, so $\mathcal{L}_{reg}$ and $\mathcal{L}_{TD^*}$ have similar targets and we expect them to have similar behaviours. Indeed, in Fig. 1 their curves mostly overlap. Second, there is a clear asymmetry between training on expectations (i.e. the learned $Q(s, a)$ or $\max_{a'} Q(s', a')$) and high-variance Monte-Carlo returns (red and blue curves in Fig. 1a). We hypothesize that since the returns $G$ are computed from the same state sequence that is used to measure the gain, $G$ is *truly* informative of the expected value of future states. Strangely, this does not seem to be the case for *past* states, which is surprising.[3] On the other hand, while $G$ appears more informative of future *expected* returns, it is not particularly more informative of future *sampled* returns than past returns, which explains the symmetric nature of the MC gain shown in Fig. 1b.

---

[3]A possible explanation is that, due to the exponential discounting nature of returns ($V(S_t) \approx \gamma^k V(S_{t+k})$ aside discontinuities when $R \neq 0$), the correlation between the current and future returns simply has a larger magnitude than with past returns. This might push DNNs to prefer to "assign capacity" w.r.t. future returns.

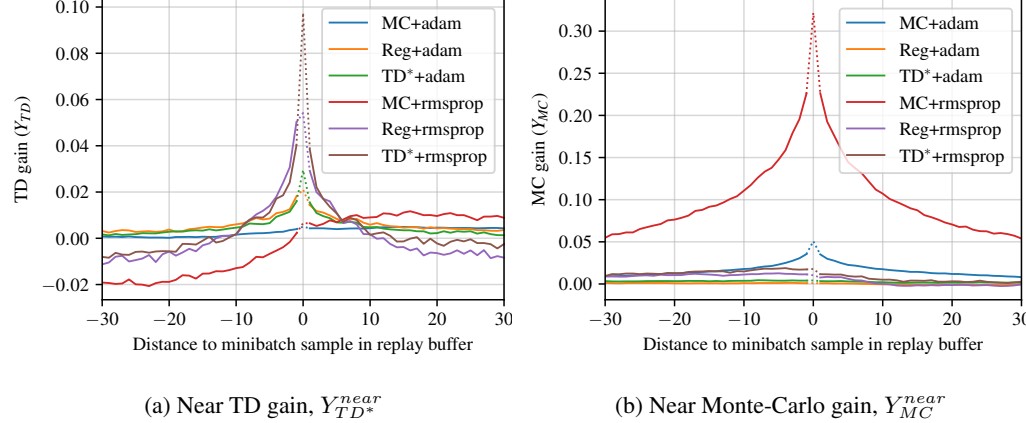

(a) Near TD gain, $Y_{TD^*}^{near}$            (b) Near Monte-Carlo gain, $Y_{MC}^{near}$

Figure 1: Supervised learning on Atari: Gain as a function of distance in the replay buffer from the update sample. We use dotted lines for the point at 0 distance, to emphasize that the corresponding state was used for the update. (a-b) The curve around 0 indicates the temporal structure captured by the TD and regression objectives.

Another striking distinction in these curves appears between the Adam (Kingma & Ba, 2015) and RMSProp (Hinton et al., 2012) optimizers.[4] When moving far away from $s$, RMSProp tends to induce a *negative* gain, while Adam tends to induce a near-zero gain. This is seen in Fig. 1a where RMSProp's TD gain is below 0 for states more than 10 steps away from the sample used in an update. Note that similar differences appear in virtually all following experiments, which we discuss later.

### 3.3 POLICY EVALUATION AND TD GAIN

We have seen that DNNs can capture some temporal structure and have good gradient update generalization when given good quality inputs *and* targets. We will now remove the expert targets generated using the pretrained $\theta^*$, but we will keep the expert inputs. This corresponds to policy evaluation on expert trajectories, and we would expect to see slightly worse generalization than in the previous case.

We run policy evaluation with 2 objectives, $\mathcal{L}_{QL}$ and $\mathcal{L}_{Sarsa}$ as defined in (5), and (6) respectively, using a frozen target to bootstrap (Mnih et al., 2013), updated after every 10k minibatches. Experiments are run on 24 Atari environments (see A.1.1) for 10 runs each. Gain results are visualized in Fig. 2, averaged over the 24 environments.

The main observation from Fig. 2a is how *narrow* the peak around 0 is, suggesting that whenever a state's value is updated, the prediction for other states does not change much in expectation, as if the representation were almost tabular, with estimates for encountered states being memorized. The conclusion we draw is that, with a fixed data distribution, DNNs bootstrapping to an evolving target network will not properly capture temporal structure, but will still be able to learn (at least in the sense of correctly approximating the value function).

Another worrying observation is that RMSProp consistently has *negative* expected gain for nearby samples (but large, larger than Adam, positive gain on $X_U$, the minibatch sample), suggesting that parameters trained with this optimizer memorize input-output pairs rather than assign capacity to generalize.

### 3.4 COMPARING MEMORIZATION BEHAVIOUR IN POLICY EVALUATION

The previous results established that *some* amount of memorization is done during TD-based policy evaluation. *Quantifying* memorization is still an open problem, but in this experiment we offer an

---

[4]It has been reported that Adam is less sensitive than RMSProp to hyperparameters in value-based methods (Hessel et al., 2018), although evidence suggests it doesn't help policy gradients (Henderson et al., 2018).

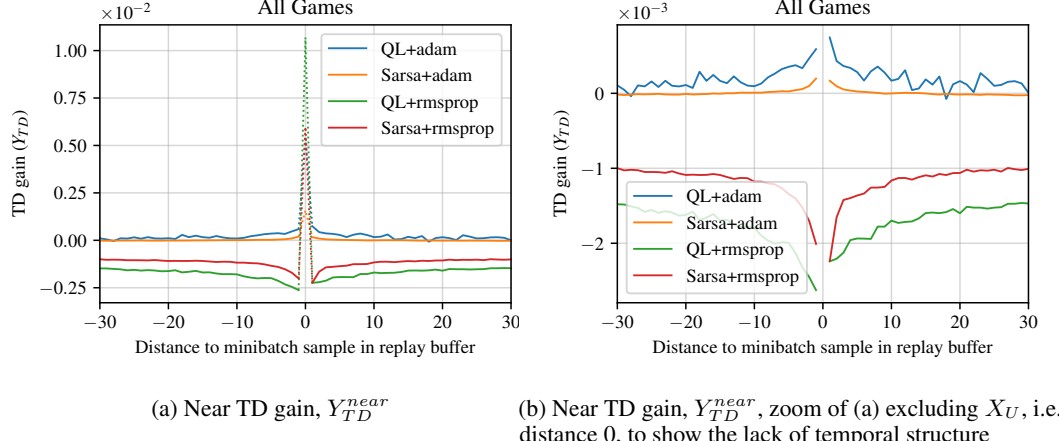

(a) Near TD gain, $Y_{TD}^{near}$

(b) Near TD gain, $Y_{TD}^{near}$, zoom of (a) excluding $X_U$, i.e. distance 0, to show the lack of temporal structure

Figure 2: Policy evaluation on Atari: Gain as a function of distance in the replay buffer of the update sample. (a) We use dotted lines for the point at 0 distance to emphasize that the corresponding state was used for the update. (a-b) Compared to regression in Fig. 1a, almost no temporal structure is captured, which can be seen by how narrow the curve is around distance 0.

interesting qualitative inspection to confirm that TD-based methods may lie somewhere between pure memorization (acting like a lookup table) and strong generalization (capturing *all* latent factors).

In Zhang et al. (2016), the authors compare images classifiers trained with true labels to classifiers trained with random labels (in which case the model *has* to simply memorize the labels), finding that, surprisingly, both can reach 0 training error. While this suggests that DNNs may also memorize when given the true labels, further studies showed many behavioural differences between the two setups, notably that DNNs first captured structure, and only afterwards fit random noise (Arpit et al., 2017).

Taking inspiration from Zhang et al. (2016), we assign a random class in $[N]$ to every state in $\mathcal{D}^*$, change our $Q$ function to be a usual classifier with $N$ outputs, and introduce a new objective, $\mathcal{L}_{rand}$, which is simply the cross-entropy between the random class and the prediction. Experiments are run on MsPacman, Breakout, and Seaquest. We use datasets of sizes 10k, 100k, and 500k, and use $N \in \{2, 10, 50\}$. Interestingly, the architecture of Mnih et al. (2013) that is reused here struggles to reach 0 error[5] (for example, a model trained with 10k samples with $N = 2$ reaches 5.7% error, while a model trained with 500k and $N = 50$ totally fails at 85% error, see Table **??**).

Fig. 3 shows the evolution during training of the distribution of $\Delta(S, A) = Q(S, A; \theta_{current}) - Q(S, A; \theta_{last(S)})$, where $\theta_{last(S)}$ represents the value of the parameters when $S$ was last used in a minibatch update, and $\theta_{current}$ represents the value of the parameters right *before* using $S$ for the most recent update. If the parameters were those of a look-up table, $\Delta$ would always be 0. For losses other than $\mathcal{L}_{rand}$ (Q-Learning, Sarsa, and MC) we reuse the results of the previous section (with a dataset size of 500k).

The difference between Fig. 3a and Fig. 3b-d is compelling, and somewhat reassuring. In Fig. 3a the log-likelihood for $\Delta = 0$ is above -2 (white) showing that it is very unlikely for the prediction at a state to have changed by more than $\pm 0.01$ when it is updated. In contrast, the distribution of $\Delta$ is more spread out in Fig. 3b-d. Combined with the fact that the memorization experiment does not reach 0 error, this allows us to confidently claim that **DQN is not fully memorizing its state space**. Even though the gain curve in Fig. 2 is very close to 0, except at the update sample (i.e. temporal structure is poorly captured), *some* structure is captured by DNNs that allow them to learn about a state without having to use it explicitly in an update.

---

[5]This could be due to the particularly shallow architecture of Mnih et al. (2013), as architectures with less parameters but more layers are commonly assumed to have more effective capacity. It has indeed been shown that deeper models can distinguish between exponentially more linear regions (Montufar et al., 2014).

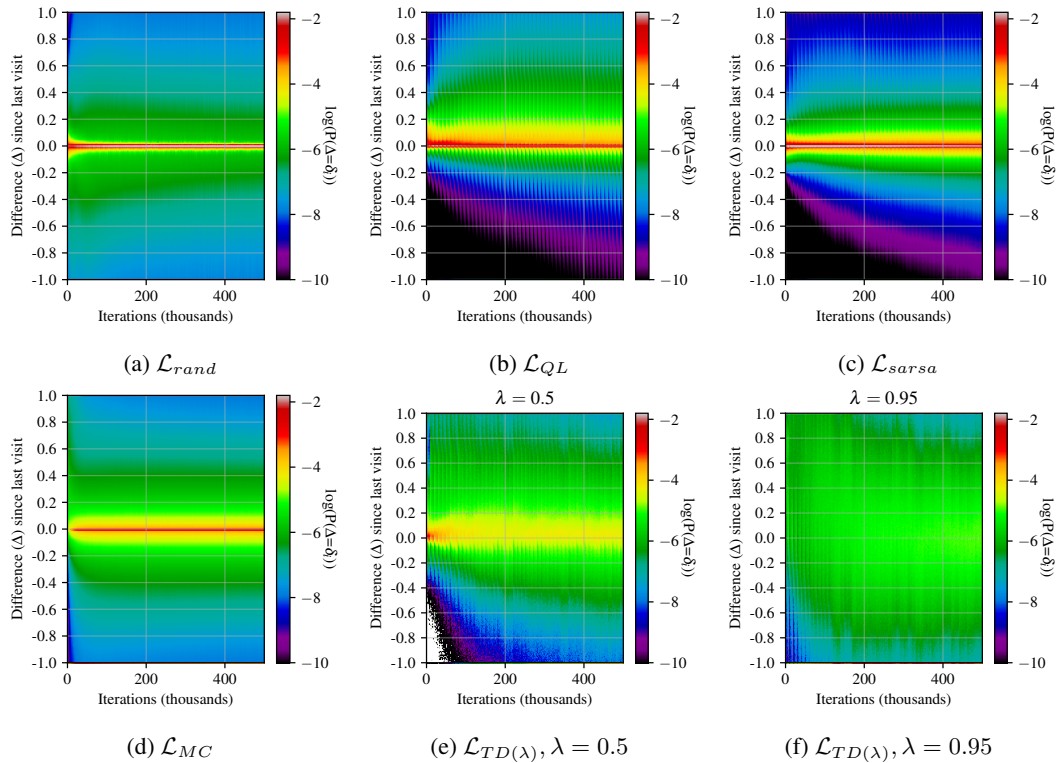

Figure 3: Policy evaluation on Atari: evolution of the distribution of $\Delta$, the difference since last visit, during training. In (a) the DNN is forced to memorize, as such the density of $\Delta$ is concentrated around 0 (thin red/white band). In (b-c), Q-Learning and Sarsa, the density is much less peaked at 0 (larger yellow/green bands) as the DNN learns about states without visiting them. In (d) the DNN learns quickly presumably without memorizing (the distribution of $\Delta$ is more spread out and not as concentrated around 0, seen by the larger yellow/green band), as it is trained on Monte-Carlo returns, and quickly converges as can be seen by the high density of positive $\Delta$s early. In (e,f) we see the effect of using $\lambda$ returns (see appendix A.6 for all values of $\lambda$).

## 3.5 TD GAIN IN CONTROL

Having removed $\theta^*$ in section 3.3, we now additionally remove $\mathcal{D}^*$ and simply perform Q-Learning from scratch on MsPacman, Asterix, and Seaquest for 10M steps.

Results are shown in Fig. 4. Interestingly, while Q-Learning does not have as strong a gain as the regressions from Fig. 1, it has a larger gain than policy evaluation. This may have several causes, and we investigate two:

- Initially, because of the random exploratory policy, the DNN sees little data, and may be able to capture a minimal set of factors of variation; then, upon seeing new states, the extracted features are forced to be mapped onto those factors of variation, improving them, leading to a natural curriculum. By looking at the *singular values* of the last hidden layer's matrix after 100k steps, we do find that there is a *consistently larger spread* in the policy evaluation case than the control case (see appendix A.3), showing that in the control case fewer factors are initially captured. This effect diminishes as training progresses.
- Having run for 10M steps, control models could have been trained on more data and thus be forced to generalize better; this turns out **not** to be the case, as measuring the same quantities for only the first 500k steps yields very similar magnitudes (see appendix A.4).

Interestingly, these results are consistent with those of Agarwal et al. (2019), who study off-policy learning. Among many other results, Agarwal et al. (2019) find that off-policy-retraining a DQN model on another DQN agent's lifetime set of trajectories yields much worse performance on average.

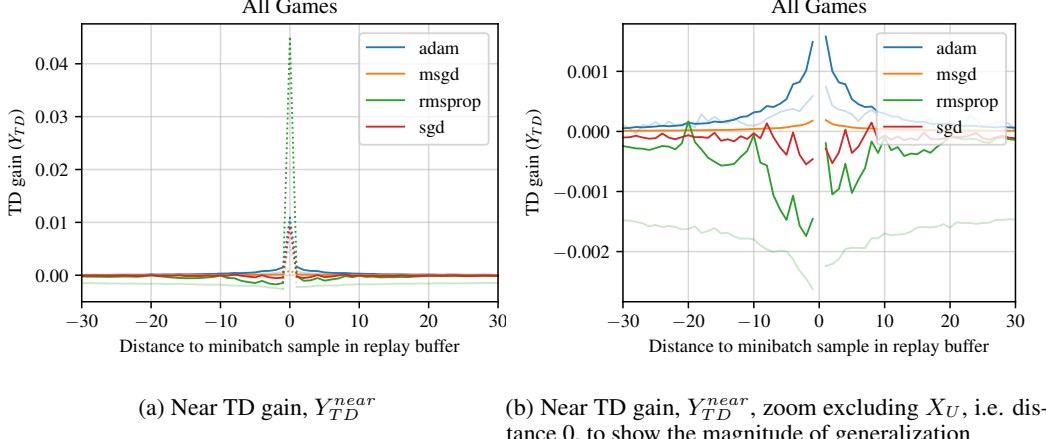

(a) Near TD gain, $Y_{TD}^{near}$

(b) Near TD gain, $Y_{TD}^{near}$, zoom excluding $X_U$, i.e. distance 0, to show the magnitude of generalization

Figure 4: Q-Learning on Atari: Gain as a function of distance in the replay buffer of the update sample. (a-b) Compared to policy evaluation, gain appears to be better, but not as large as for regression. (b) For Adam and RMSProp, we include the corresponding curves for policy evaluation in lighter shades.

While the authors suggest the strongly off-policy aspect of this experiment as the cause, our results still show differences between control-Q-Learning and policy-evaluation-Q-Learning, which are both done "on-policy" in our setup, suggesting there are more factors at play than only off-policyness.

Note that we also additionally run experiments with SGD and Momentum-SGD optimizers to highlight the difference between Adam, that has a momentum component, and RMSprop, which only scales per-parameter learning rates. Predictably, Momentum-SGD's behaviour is similar to Adam, and SGD's to RMSprop.

## 3.6 TD($\lambda$) AND RELIANCE ON BOOTSTRAPPING

TD($\lambda$) trades off between the immediate biased estimates of the future values and the true return through its $\lambda$ parameter. To observe the effect of this parameter we perform policy evaluation on $\mathcal{D}^*$ with the $\mathcal{L}_{TD(\lambda)}$ objective on MsPacman.

Results are shown in Fig. 5, where we can observe that (1) increasing $\lambda$ increases near gain without overly increasing update-sample gain (2) as for $\mathcal{L}_{MC}$, there is an asymmetry: updating informs us more about the future than about the past, on average. Results for the distribution of $\Delta$ are shown in Fig. 3(e,f) (and appendix A.6), where we see that the closer $\lambda$ is to 1, the more the TD($\lambda$) objective creates updates that affect all states.

These results seem to indicate that TD($\lambda$) better captures factors of variation. One cause could be that the more one relies on a sequence of DNN predictions (i.e. the sequence of $n$-step returns of the $\lambda$-return depend on the successive $V(S_{t+i})$) to build a target, the more correlation there is between states and targets (due to DNN smoothness), and the more temporal coherence there is (and thus more opportunities for DNNs to capture the temporal dimension's correlations). This is hard to verify empirically, but we can proxy the correlation measure via the similarity between gradients. We do indeed find that the closer $\lambda$ is to 1, the higher the average cosine similarity between gradients is (see appendix A.5). This suggests that it may be advantageous to use $\lambda$-returns in environments where generalization is important.

## 3.7 TESTING GENERALIZATION WITH AN INTRA-TASK TEST SET

Another way to assess whether agents fail to properly generalize in the sense of statistical inference – *making predictions* about states without visiting them – is to create a test set to measure generalization error. We do so on the MsPacman Atari environment, as it contains many opportunities for generalization in translational invariances (locally, the optimal action only depends on the surrounding

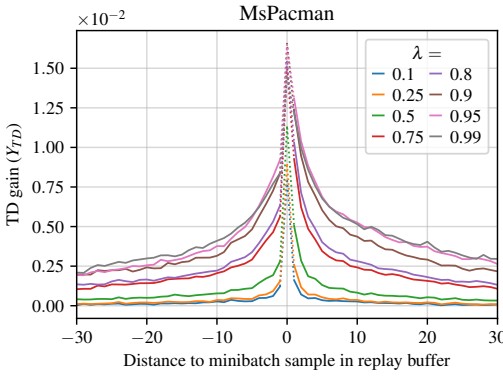
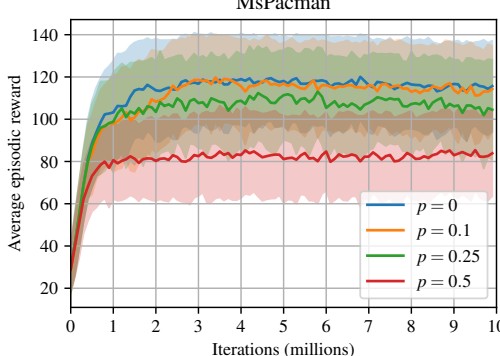

Figure 5: TD gain for policy evaluation with TD($\lambda$) & Adam. Note the larger gain as $\lambda$ goes to 1, as well as the asymmetry around 0.

Figure 6: Episodic rewards over Q-Learning on Atari. We train an agent while witholding states from the training set with probability $p$.

configuration of the agent, reward pellets, ghosts). We train our agent with the usual DQN setup (Mnih et al., 2013) but prevent the insertion of a state into the replay buffer with some probability $p$. More specifically, we use the RAM (ground truth) state information to exclude observations from training. We run 5 seeds for each $p \in \{0, 0.1, 0.25, 0.5\}$.

Results are shown in Fig. 6, where we see that witholding only 10% of states already slightly affects agents. At 50%, performance is significantly reduced. While this is somewhat expected and consistent with the literature (Farebrother et al., 2018), it again attests that TD-based methods can struggle with generalization, as observed also by Packer et al. (2018), who study interpolation and extrapolation failures in deep RL agents.

### 3.8 ADDITIONAL OBSERVATIONS

**On other structures** Our figures mostly show gradient update generalization gain as a function of "time" (temporal distance within a trajectory), but there might be structure elsewhere. We measured gain as a function of 3 different metrics: ground truth state distance by reusing the Annotated Atari RAM of Anand et al. (2019), value distance (as DNNs may alias states with the same value), and feature distance. Unfortunately, we were unable to find correlations (see appendix A.2).

**On convergence** Figures 1, 2, and 4 show values averaged over the course of training. We find that except in the first few iterations, these curves remain constant throughout training (see figures in A.4) and show no sign of convergence. This is also consistent with previous studies, as DQN is known to not converge on Atari (Anschel et al., 2017).

**On variance** While $Var(Y_{\mathcal{L}})$ tends to be large, we find that the confidence interval of the *mean* is always small, and would barely appear on most of our plots. Additionally, although generalization gain is typically a fraction of the magnitude of the value function, it is consistently non-zero.

**On optimizers** We find that the systematic differences we see between Adam and RMSProp also occur in behaviour, where control agents trained with RMSProp tend to get slightly more reward. An interpretation of our results is that RMSProp memorizes faster than Adam: it has much larger on-sample gain, it tends to make the singular values of the weight matrices larger, and it has negative near-sample gain, suggesting that capacity is spent memorizing on average. In Atari tasks, memorization can be an efficient strategy (although it is sensitive to noise, see Machado et al. (2018)). Hence, the better performance of RMSProp *on Atari* is consistent with our claims. This property may not be as desireable in more complex environments requiring generalization.

## 4 DISCUSSION

RL is generally considered a harder problem than supervised learning. Hence, the fact that TD-style methods require more samples than supervised learning when used with deep nets is not necessarily

surprising. However, with the same data and the same final targets (the "true" value function), it is not clear why TD updates lead to parameters that generalize worse than supervised learning. This could be a problem, as most RL methods rely on the TD mechanism in one way or another. In particular, our results show that both Q-Learning and Sarsa generalize poorly, leading to DNNs that memorize the training data (not unlike table lookup). Our results also suggest that TD($\lambda$), although not widely used in recent DRL, improves generalization. Finally, we find differences between Adam and RMSProp that we initially did not anticipate. Very little work has been done to understand and improve the coupling between optimizers and TD, and our results indicate that this would be an important future work direction.

Our work suggests that the RL community should pay special attention to the current research on generalization in DNNs, because approaching the TD bootstrapping mechanism as a supervised learning problem does not seem to leverage the full generalization potential of DNNs.

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

## A  APPENDIX

### A.1  EXTRA EXPERIMENTAL DETAILS

#### A.1.1  ATARI GAMES

Some sections are only run with 3 Atari games, MsPacman, Asterix and Seaquest. We chose these three games as they exhibit features that seem particularly amenable to generalization.

The full 24 games we use for policy evaluation tests are Alien, Amidar, Assault, Asterix, BankHeist, Boxing, Breakout, ChopperCommand, CrazyClimber, DemonAttack, Freeway, Frostbite, Gopher, Hero, Jamesbond, Kangaroo, Krull, KungFuMaster, MsPacman, PrivateEye, Qbert, RoadRunner, Seaquest, and UpNDown.

All our experiments train parameters with the following architecture, as per Mnih et al. (2013): 3 convolutional layers with kernels of shape $4 \times 32 \times 8 \times 8$, $32 \times 64 \times 4 \times 4$, and $64 \times 64 \times 3 \times 3$ and with stride 4, 2, and 1 respectively, followed by two fully-connected layers of shape $9216 \times 512$ and $512 \times |\mathcal{A}|$, $\mathcal{A}$ being the legal action set for a given game. All activation are leaky ReLUs (Maas et al.) except for the last layer which is linear (as it outputs value functions or unnormalized logits for the classification experiment).

| $|\mathcal{D}| \setminus N$ | 2 | 10 | 50 |
|---|---|---|---|
| 10k | 5.7 | 8.7 | 8.5 |
| 100k | 38.8 | 45.6 | 56.5 |
| 500k | 44.7 | 79.3 | 85.8 |

Table 1: Classification error (%) for memorization experiment of Section 3.4. $|\mathcal{D}|$ is the dataset size. $N$ is the number of classes. Classes are uniformly randomly assigned to states.

#### A.1.2  FIGURE HYPERPARAMETERS

The experiments of Figures 1 and 2 run for 500k steps, measuring gains every 500 updates. A learning rate of $10^{-4}$ is used, with L2 weight regularization of $10^{-4}$. When boostrapping to a frozen network, the frozen network is updated every 10k updates. We use $\gamma = 0.99$, a minibatch size of 32, an $\epsilon$ of 5% to generate $\mathcal{D}^*$, and a buffer size of 500k. Each setting is run with 10 random seeds. The random seeds affect the generation of $\mathcal{D}^*$, the weight initialization, the minibatch sampling, and the choice of actions in $\epsilon$-greedy rollouts.

The values of $\Delta$ in Figure 3 are measured at every update. The memorization procedure is also run for 500k steps, which is enough for the model to seemingly converge (albeit to a non 0 error).

The Q-Learning control experiments of Figures 4 and 6 are run for 10M steps with the same setup as previously described. Each setting is run with 10 random seeds. For every environment step, one minibatch update is done.

The experiments of Figure 5 are run for 500k steps, as previously described. $\lambda$-targets are computed with the forward view, using the frozen network to compute the target values – this allows us to cheaply recompute all $\lambda$-targets once every 10k steps when we update the frozen network. Each setting is run with 5 random seeds.

Note that we did experiment with various learning rates, momentum settings, DNN regularizations, and other common tricks–our conclusions remained the same, but for simplicity of presentation we stick to commonly used hyperparameters.

#### A.1.3  MINIBATCHES

All plots are generated from measures taken during training, either during policy evaluation or Q-Learning. While learning rates and minibatch sizes do have an influence on the results, most of the time results remain mostly the same and for simplicity of presentation only a subset of the experiments we performed are shown. Ideally, gain should be measured for a single example, but we found that results were the same and lower variance for larger minibatch sizes. As such we consistently use 32 examples per minibatch, which reflects current practice in RL.

### A.1.4 REPRODUCIBILITY CHECKLIST

We follow the Machine Learning reproducibility checklist (Pineau, 2019), and refer to corresponding sections in the text when relevant.

For all models and algorithms presented, check if you include:

- **A clear description of the mathematical setting, algorithm, and/or model.** We use unmodified algorithms, described in the technical background, and only analyse their behaviour. The measures we propose are straightforward to implement and only require minimal changes
- **An analysis of the complexity (time, space, sample size) of any algorithm.** The measures we propose only add a constant instrumentation overhead.
- **A link to a downloadable source code, with specification of all dependencies, including external libraries.** All code is included in supplementary materials, dependencies are documented within.

For any theoretical claim, check if you include:

- **A statement of the result.** We make no theoretical claim.
- **A clear explanation of any assumptions.** idem.
- **A complete proof of the claim.** idem.

For all figures and tables that present empirical results, check if you include:

- **A complete description of the data collection process, including sample size.** We collect data by running standard implementations of common algorithms with repeated runs.
- **A link to a downloadable version of the dataset or simulation environment.** Included in the code available in supplementary materials.
- **An explanation of any data that were excluded, description of any pre-processing step.** We generally chose hyperparameters that best represent state-of-the-art usage, then if necessary that best represent our findings. In most cases only minor learning rate adjutments were necessary, although they would not significantly change most plots.
- **An explanation of how samples were allocated for training / validation / testing.** As we are only interested in the training process this is not fully applicable.
- **The range of hyper-parameters considered, method to select the best hyper-parameter configuration, and specification of all hyper-parameters used to generate results.** See section A.1.
- **The exact number of evaluation runs.** idem.
- **A description of how experiments were run.** idem.
- **A clear definition of the specific measure or statistics used to report results.** See section 3.
- **Clearly defined error bars.** Figures with error bars compute a bootstrapped 95% confidence interval of the mean.
- **A description of results with central tendency(e.g. mean) & variation(e.g. stddev).** idem.
- **A description of the computing infrastructure used** Almost all experiments were run on P100 GPUs, otherwise they were run on Intel i7 processors.

### A.2 LOOKING FOR OTHER STRUCTURES IN TD GAIN

On top of measuring $Y_{TD}^{near}$, i.e. TD gain for nearby examples, we also measure TD gain as a function of ground truth distance, value distance and feature distance. In all cases we find the expected gain to be close to 0 with only few interesting patterns to the gain curves.

**Ground truth distance** The Arcade Learning Environment (Bellemare et al., 2013) provides, in addition to pixel observations, the current memory (RAM) state of the game, which consists of 128 8-bit values. In Anand et al. (2019), the authors annotate the relevant individual bytes, discarding those not representing anything of value. For example, in MsPacman they identify 17 values consisting of the player's position, the ghosts' positions, etc. In Figure 7a, we plot $Y_{TD}$ as a function of the L1 RAM distance (for only the relevant bytes) between $X_U$ and $X$ (i.e. the update samples and the test samples; we use 2048 test samples each time we perform this measure, which is every 1000 minibatch updates, for a total of 1024000 samples). There seems to be no correlation between $Y_{TD}$ and the distance.

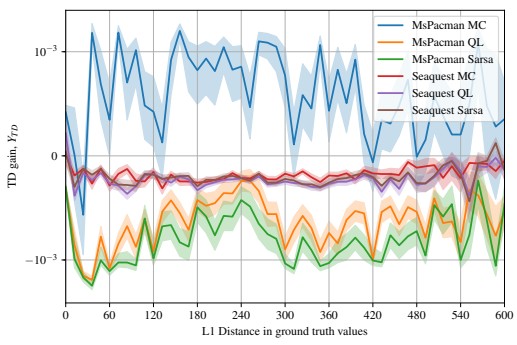 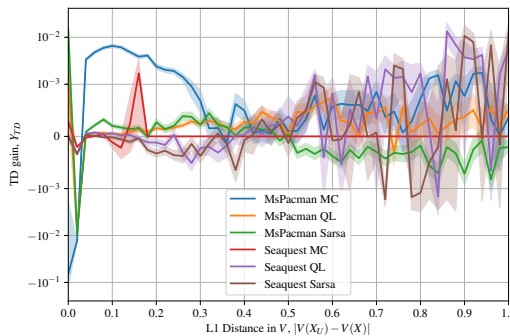

(a) Policy evaluation, TD gain as a function of anno-tated ground truth distance.

(b) Policy evaluation, TD gain as a function of distance in value predictions.

Figure 7: TD gain as a function of other metrics. Shaded regions are the standard error to the mean.

**Value distance** DNNs might alias together (blend into a single point in latent space) all states that have the same value. This would mean that changing the value prediction for one of those states will change the value prediction for all other aliased states. In Figure 7b we see that this effect might occur in the $[0, 0.02]$ bin, where TD gain tends to be positive for TD methods. For Monte-Carlo returns in MsPacman, the effect is negative on average in the $[0, 0.02]$ bin, but otherwise positive from $[0.02, 0.3]$. There seems to be no correlation otherwise to TD methods.

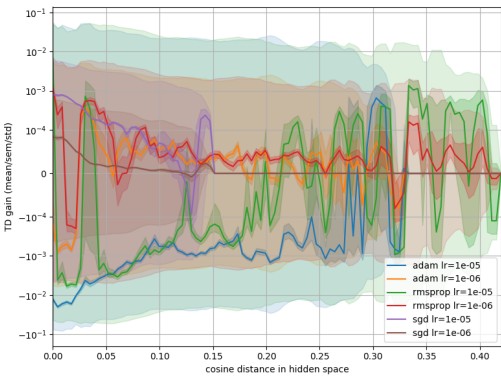

Figure 8: Policy evaluation with $\mathcal{L}_{QL}$, TD gain as a function of cosine distance in hidden space (second to last layer).

**Feature distance** We perform again a similar experiment on MsPacman, but using as a distance metric the cosine similarity of activations in the second to last layer. While we do find a minor correlation with distance, its magnitude is very small. In addition, this correlation is easily explained by the geometry of the shallow neural network we use. Since we use leaky ReLUs and measure distance in the second to last layer, when any two states are close in the hidden space, changing the mapping to the output for one state will change the output for the other state with high probability. Thus what we observe here is closer to how much states are entangled in the hidden space rather than a real measure of how update generalization is linked to distance (in true or hidden space). Interestingly, hidden states trained with SGD seem much more entangled, their cosine distance is often 0. This does not seem to be the case with networks trained with Adam and RMSProp.

## A.3 SINGULAR VALUES, CONTROL VS POLICY EVALUATION

Figure 9 shows the spread of singular values after 100k minibatch updates on MsPacman for the Q-Learning objective and Adam/RMSProp. The difference between the control case and policy evaluation supports our hypothesis that policy evaluation initially captures more factors of variation.

It remains unclear if the effect of the control case initially having fewer captured factors of variation leads to a form of feature curriculum.

Figure 10 shows the spread of singular values after 500k minibatch updates for TD($\lambda$). Interestingly, larger $\lambda$ values yield larger singular values and a wider distribution. Presumably, TD($\lambda$) having a less biased objective allows the parameters to capture all the factors of variation faster rather than to rely on bootstrapping to gradually learn them.

Note that current literature suggests that having fewer large singular values is a sign of generalization *in classifiers*, see in particular Oymak et al. (2019), as well as Morcos et al. (2018) and Raghu et al. (2017). It is not clear whether this holds for regression, nor in our case for regression to value functions, but interestingly all runs (even for TD($\lambda$)), have a dramatic cutoff in singular values after about the 200th SV, suggesting that there may be in this order of magnitude many underlying factors in MsPacman, and that by changing the objective and the data distribution, a DNN may be able to capture them faster or slower.

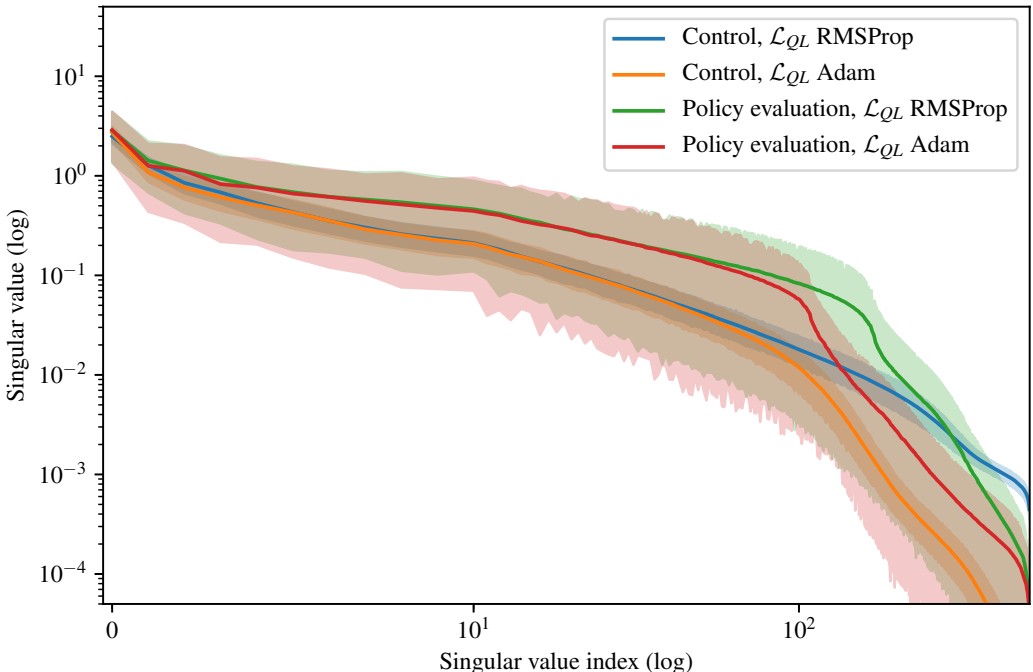

Figure 9: Spread of singular values after 100k iterations. Despite having seen roughly the same amount of data, the control experiments generally have seen fewer unique states, which may explain the observed difference. Shaded regions show bootstrapped 95% confidence intervals.

### A.4   EVOLUTION OF TD GAIN WITH TRAINING

Figure 11 shows the evolution of TD gain during training; in relation to previous figures like Figure 1, the $y$ axis is now Fig. 1's $x$ axis – the distance to the update sample in the replay buffer, the $y$ axis is now training time, and the color is now Fig. 1's $y$ axis – the magnitude of the TD gain.

Another interesting observation with respect to the evluation of TD gain from Fig. 3 is that the density is asymmetric, predictions tend to increase during training. This is consistent with the fact that DNNs are initialized to predict 0 in expectation, while value functions for agents that receive mostly positive rewards will tend to be positive. Also note that Fig. 3 differs very little when using Adam over RMSProp.

### A.5   COSINE SIMILARITIES IN TD($\lambda$)

Figures 12 and 13 show cosine similarities between gradients after 500k updates.

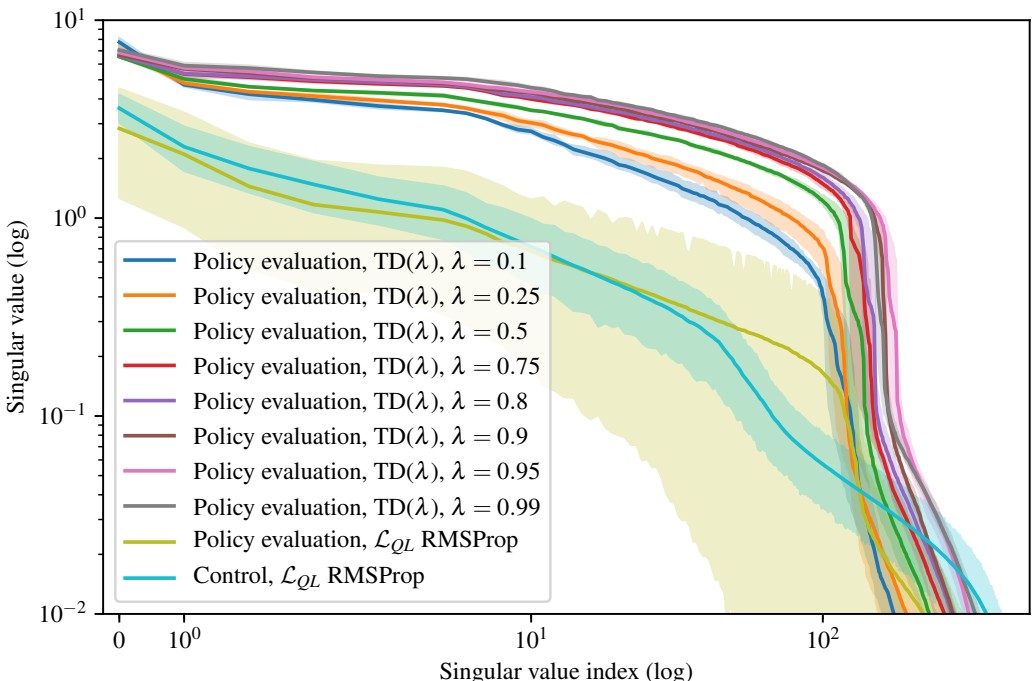

Figure 10: Spread of singular values after 500k iterations. Shaded regions show bootstrapped 95% confidence intervals.

## A.6 DIFFERENCES SINCE LAST VISIT IN TD($\lambda$)

See Figure 14, and Figure 3 for reference.

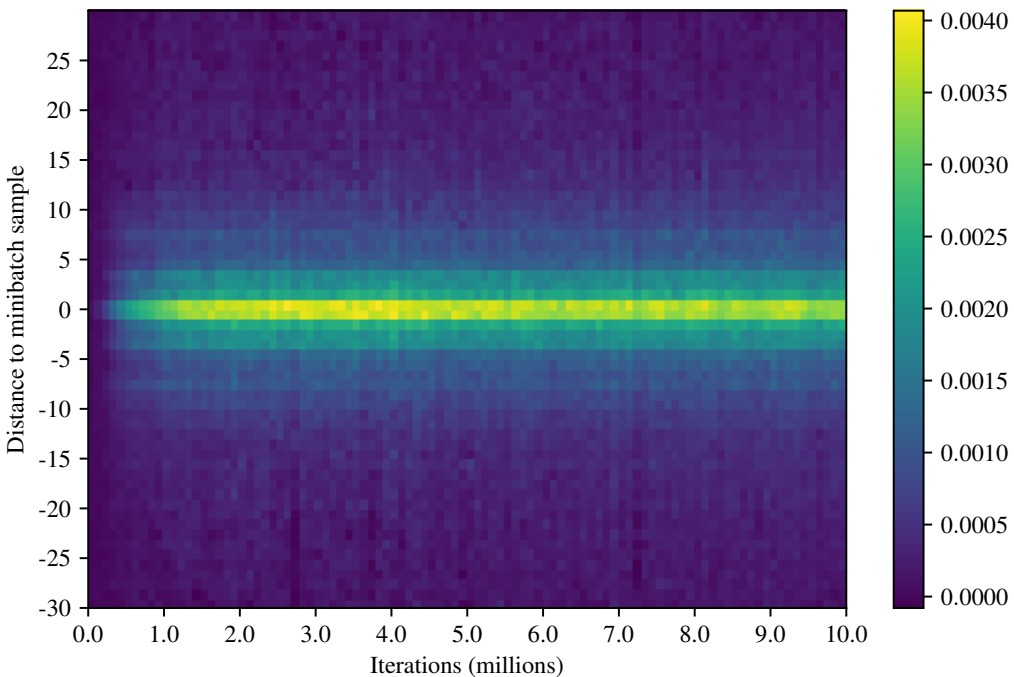

Figure 11: Evolution of TD gain, $Y_{TD}^{near}$ during training. Control experiment with Adam, MsPacman, averaged over 10 runs. Note that index 0 is excluded as its magnitude would be too large and dim all other values.

### A.7    Is Near TD gain indicative of speed of learning?

In Figure 15 we plot, for an agent trained on MsPacman, the lifetime agent reward (i.e. the AUC of a curve as in Figure 6) as a function of the lifetime average near TD gain $Y_{TD}^{near}$. The line is a linear regression of the points, with a correlation coefficient of $r = 0.4337$. We vary the capacity of the agent by changing the number of hidden units of every layer, between $0.25\times$ and $4 times$ the original size.

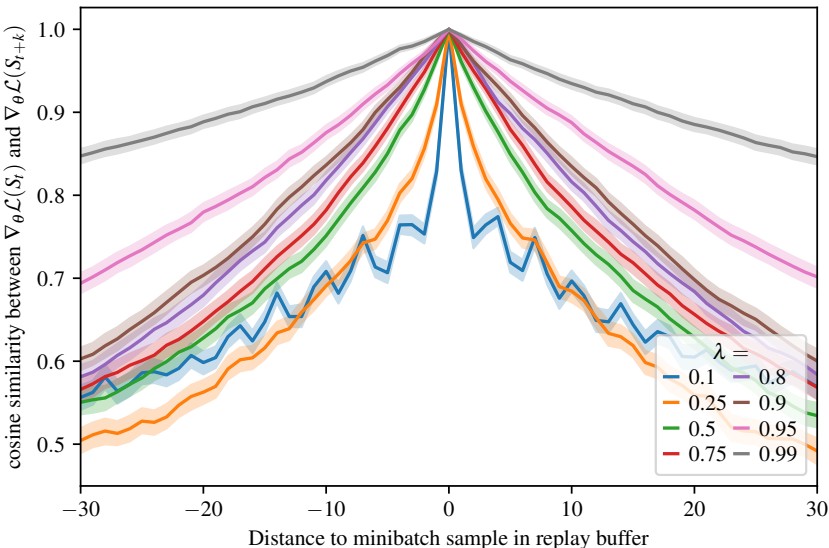

Figure 12: Measuring the cosine similarity in a sequence. The $y$ axis shows the cosine similarity between $\nabla_\theta \mathcal{L}_{TD(\lambda)}(S_t)$ and $\nabla_\theta \mathcal{L}_{TD(\lambda)}(S_{t+k})$ for the nearest $k \in [-30, 30]$ neighbours in the replay buffer. The larger $\lambda$ is, the more similar gradient updates are for two states when those states are close in time. Shaded regions show bootstrapped 95% confidence intervals.

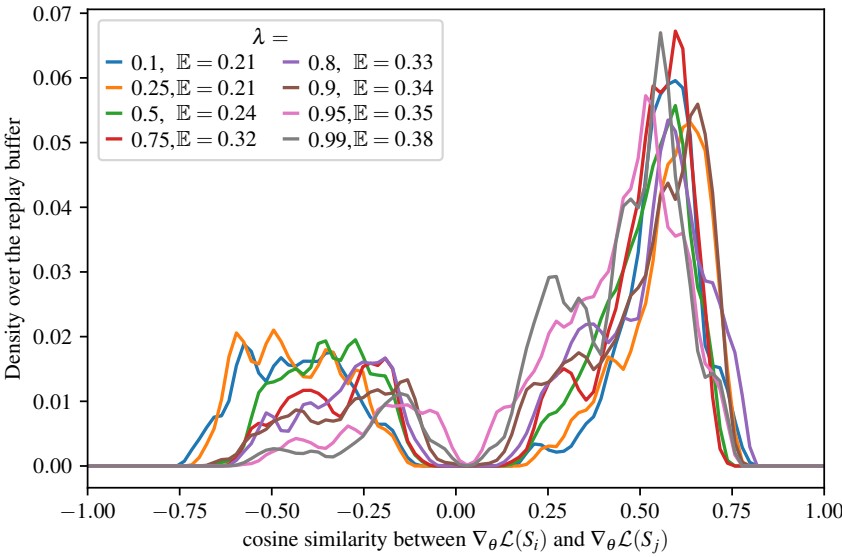

Figure 13: Measuring the distribution of cosine similarity. The $y$ axis shows the density of a given cosine similarity ($x$ axis) between $\nabla_\theta \mathcal{L}_{TD(\lambda)}(S_i)$ and $\nabla_\theta \mathcal{L}_{TD(\lambda)}(S_j)$ for 20k $(i, j)$ pairs in the replay buffer. The distributions for *positive* cosine similarities are similar for all values of $\lambda$. On the other hand, on the left side of the plot we can see that when $\lambda$ gets closer to 0, there are more and more gradients "pointing the wrong way", i.e. with a negative cosine similarity. The labels show the expected cosine similarity over the replay buffer ($\mathbb{E} = ...$), which show that as $\lambda$ increases, gradients are more similar. Interestingly there are almost no pairs of states with a cosine similarity of 0.

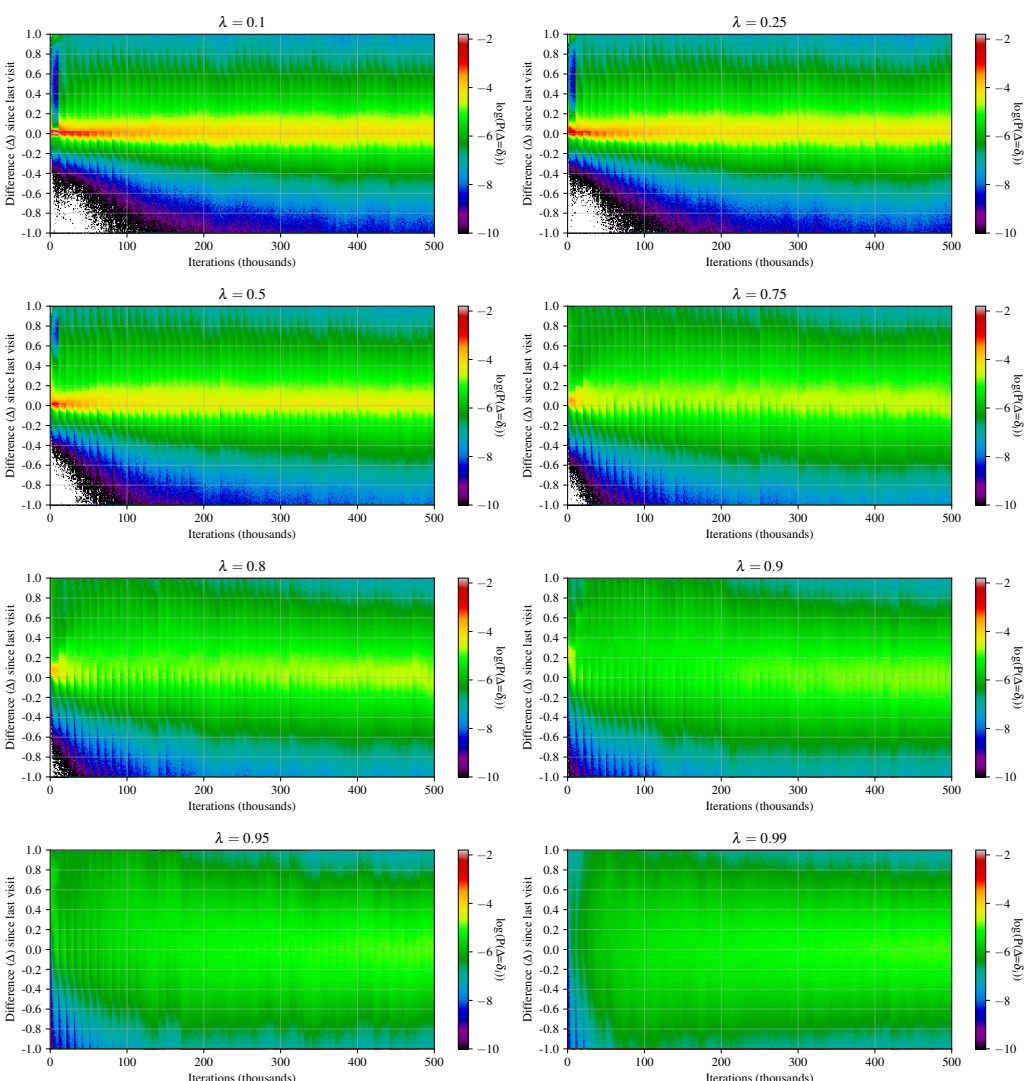

Figure 14: Reproducing Fig. 3 but for all values of $\lambda$ that were tested.

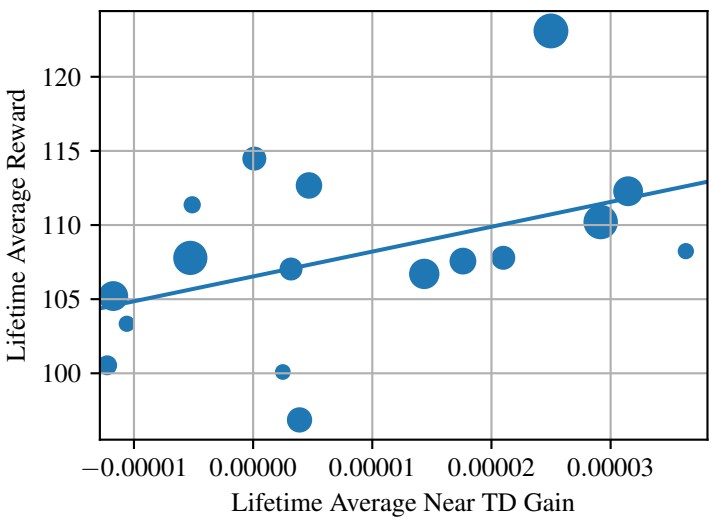

Figure 15: The lifetime agent reward (i.e. the AUC of a curve as in Figure 6) as a function of the lifetime average near TD gain $Y_{TD}^{near}$. The line is a linear regression of the points, with a correlation coefficient of $r = 0.4337$. We vary the capacity of the agent by changing the number of hidden units of every layer, which is reflected by the size of the circles.

