# OpenReview forum: "Assessing Generalization in TD methods for Deep Reinforcement Learning"
_ICLR.cc/2020/Conference — Reject_

### Official Review · AnonReviewer1 · 2019-10-21
**Official Blind Review #1**

**Rating:** 3

**Review:**

Summary:
This paper performs an empirical evaluation of generalization by TD methods with neural nets as function approximators. To quantify generalization, the paper considers the change in the loss function at similar states to the one where the update rule is being applied (where “similar” is usually defined as nearby in time). It comes to a variety of conclusions including that TD(0) does not induce much generalization, that TD(0) does not induce as much generalization as supervised learning, and that the choice of optimizer and objective changes the behavior according to their generalization criteria in various ways.


Decision:
This paper should be rejected because (1) the motivation is unsubstantiated, (2) the main metrics used (“gain” at nearby states and difference since last visit) are of questionable importance, and (3) the conclusions are often vague and not informative.


Main argument:
Motivation:
- The paper motivates the need for an evaluation of “gradient update generalization” by claiming that it is related to sample complexity and brittleness of RL algorithms. While I agree that this is plausible, there is nothing empirical or theoretical to support this claim in the paper or in the references. This is a significant problem since this assumed connection underlies everything in the paper.
- Also, it is this sort of generalization that differentiates the function approximation setting (where there are no convergence guarantees for TD without strong assumptions) from the tabular setting (where there are convergence and sample complexity guarantees).

Metrics:
- The main metric used is TD gain on temporally nearby states. The TD gain is defined as the reduction in the squared TD error at some state s’ when an update is applied at some other state s. Note this metric does not capture all update generalization, but only update generalization as it effects the TD error.
- It is not evident, nor supported by the paper, that improvements in this metric at nearby states necessarily improve performance of the algorithm. This is especially true when there is a tradeoff between improvement at very nearby vs. somewhat nearby states, it is not clear which behavior is preferable (and this behavior seems to occur in experiments). As a result there is no clear way to use this metric to determine which algorithms are preferable.
- The other metric used in the paper is the change in value function at a state since the last time that state was sampled from the buffer. It is also not clear whether this measurement is necessarily important for the same reasons as above.

Conclusions:
- In general the results are presented in plots that do not give clear implications and are difficult to read. I understand that we cannot expect completely clean results on such empirical questions, but the results would be potentially much more convincing and clear if the hypotheses were clearly stated and then one plot could summarize the result with respect to each hypothesis. For example, the gain plots are difficult to compare and interpret clearly and the memorization plots do not give such clear results (eg. 3a and 3d look visually fairly similar). Another example is the comparisons between optimizers, which are fine to have as a specific point in one section, but do not need to be in every plot.
- The main result claimed in the paper is that there is little generalization from TD(0) when compared to supervised learning. This seems to be born out by the difference between figure 1a and 2a, but the difference in scale makes it a bit difficult. It is also not clear how to quantify this result or what the implications are.
- The plots are averaged across all states over all of training and all environments, while this is somewhat rationalized in figure 11, I am worried that this may be covering some additional complexity/ambiguity in the results.
- The result about TD(lambda) seems to be born out by figures 3e and f and 5, but is also unsurprising since the objective explicitly averages across temporally nearby states. Again it is not clear how this temporal consistency of updates should be interpreted in terms of the goals of the RL algorithms.

At a higher level, the paper feels like a solid preliminary set of experiments rather than a paper organized around a clear motivating idea with clear hypotheses to test. The results would become more interesting if the metrics used could be connected back (either empirically or theoretically) to an objective. For example, does generalization in the sense defined in the paper give better performance at value estimation or return maximization? Can these results be quantified in a more direct way than the plots presented in the paper?


Additional feedback:
- The plots all have different scales which makes them difficult to compare
- Using “distance” to refer to the relative tilmestep of samples in the replay buffer is confusing (distance cannot be negative)
- There are strange visual artifacts (horizontal lines) in Figures 3, 5, 10, 12, 13, and 14
- Sections 3.7 and 3.8 very briefly present results that seem to distract from the main thread of the paper
- Using one network architecture across all experiments seems like it may have a significant impact on the results. I understand that the architecture chosen is standard and testing different architectures is time consuming, but making broad claims about the algorithms based on only one architecture is potentially dangerous.
- I assume that the replay buffer is always being sampled uniformly, but I could not find this detail in the paper.

**Experience Assessment:**

I have published one or two papers in this area.

**Review Assessment: Checking Correctness Of Derivations And Theory:**

N/A

**Review Assessment: Checking Correctness Of Experiments:**

I assessed the sensibility of the experiments.

**Review Assessment: Thoroughness In Paper Reading:**

I read the paper at least twice and used my best judgement in assessing the paper.

---

> ### Author Response · Authors · 2019-11-11
> **Feedback**
>
> Dear reviewer, thank you for your feedback. The nuance of the phenomenon we are attempting to characterize does bring challenges, and requires simultaneously considering new questions, new experiments, and new metrics.  We welcome your suggestions for developing and presenting findings in a clearer manner.
>
> > On (1) and (2), motivation and importance of the metric
> A previous version of this paper had a scatter plot showing the link between the average magnitude of update gain and lifetime rewards of an agent. It appeared obvious to us that this link existed, but we will reintroduce this plot, which is empirical evidence of $Y^{near}$ being indicative of speed of learning (correlation coefficient of r=0.433, see Fig 15 in revised version).
>
> > On (3), vague and uninformative conclusions
> We believe we have identified a behaviour which, at least to us and most RL researchers with whom we discussed this, is surprising and unexpected. We have attempted in many ways to understand how this behaviour arises, ruling out many hypotheses, unfortunately without much success so far. Knowing its cause would of course bring even greater insight, but we nonetheless strongly believe that publishing this unexpected result would be highly valuable for the community.
> We disagree that these results are uninformative. Most papers have a conclusion along the lines of “do this” or “don’t do that”; our paper instead suggests that the Deep RL community may be unaware of a problem with large ramifications. It is our opinion that identifying directions of research is valuable.
>
> > On difference since last visit
> A common observation of memorizing deep neural networks is that they act like nearest-neighbour classifiers or even tabular lookups. This measure simply aims to disprove this for DQN by showing that, unlike for a DNN trained to memorize, states change value without being “visited” for a gradient update. So while the first results suggest that DQN could just be “memorizing” states, these results suggest that DQN doesn’t 100% memorize.
>
> > On averaging plots over states and environments
> We tried many different plots to see if patterns would emerge. For example, we tried separating curves by how far the next reward was, or by distance to a terminal state. We were unable to see any robust pattern specific to some characterization of the MDP/trajectories.
> For environments, there is some amount of difference, usually depending on how dense rewards are, but these differences did not strongly affect the shape nor the magnitude of the update gain curves. As such, we simply averaged over all environments.
>
> > On plot scale
> We agree that this should be clearer in the text. Figure 1 and Figure 2’s y axes are an order of magnitude apart (Figure 2a’s y axis is 10x smaller, Figure 2b’s 100x). Showing them both in the same plot is feasible but it seemed to us more informative to separate the two.
>
> > On using “distance”
> You are right, we will replace “distance” with “offset”.
>
> > On visual artifacts
> We are sorry to hear this. We are unable to reproduce this problem, but will investigate on other browsers/OS. Here is a rasterized version of the pdf if you need it: https://imgur.com/a/Qdv0JLt
>
> > Architectures
> As mentioned previously we will add a scatter plot showing the effect of varying the architecture and capacity.
>
> > Replay buffer
> Yes, the buffer is sampled uniformly. We also briefly experimented with Prioritized Experience Replay, which had slightly larger update gain but still in the same order of magnitude.

---

> > ### Comment · AnonReviewer1 · 2019-11-13
> > **re: Feedback**
> >
> > Points (1) and (2):
> > I do not think this plot is a sufficient response to change my opinion. To address the plot specifically, the correlation is quite low, if r = .433 then the more common r^2 is .187. This seems like a low correlation to stake the main assumption of the paper on. Even then it does not make sense to me why each data point corresponds to a different network capacity, why should this be the varied quantity to generate different datapoints for this trend?
> > Besides this plot, I think the main point of the review went unaddressed that there is no clear and formal rationale for studying the proposed metrics.
> >
> > An additional worry:
> > Going back through the paper I had an additional concern with the results. The gain plots all have quite different values at the 0-offset datapoint. Under the definition of the gain, this just depends on the step size and steepness of the loss at that point, nothing about generalization. It seems that this should be normalized to be able to compare generalization results across different settings and may actually explain much of the observed variation in the plots.
> >
> > Point (3):
> > I agree that raising interesting new problems is an important part of research, but I do not think that this paper has done so. To me there is neither a clear and concise statement of an open problem nor sufficiently clear experimental evidence that there is a fundamental problem being raised by the paper.  As I said in the review, there are some interesting preliminary results, but to motivate future research the presentation needs to be cleaned up substantially.

---

> > > ### Author Response · Authors · 2019-11-14
> > > **re: Feedback**
> > >
> > > (1&2)
> > > Even if no quantity is varied but the random seed of the experiment, this trend still exists. It also seems to hold across RL methods; as mentioned we tried PER, which had slightly better gain and also rewards; we also briefly tried C51, which seemed to also have better gains and rewards. Are there other quantities that you suggest we should try varying?
> > > Of course, the gain is a high variance measure. A number of other things contribute to the final agent’s performance. This gain measure does not perfectly explain performance, that would be _very_ surprising, but it certainly looks like there is some link.
> > > The case we make is *not* that researchers should be using this measure to choose their hyperparameters, it is simply an observational one: we know that DNNs on supervised learning (SL) tasks have a certain sets of behaviours, but we find that some of these behaviours are lacking in Deep RL. All the differences we observe are consistent with many other papers, including common intuitions in SL with DNNs.
> > >
> > > An important rationale for studying this metric is to be able to compare SL and RL, not even necessarily RL algorithms together. Considering we do find major differences between SL and RL using this metric, it seems logical that something non-trivial is causing these differences. Considering that after some decent amount of investigating we are still unable to identify these causes, it seems natural to share this problem with the community.
> > >
> > > Another important rationale for this metric, mentioned in the paper, is that “any kind of generalization must arise through the accumulation of parameter updates”. We should expect something about these parameter updates that makes them “general”. This metric is the most successful measure of such a phenomenon we could find.
> > >
> > > One of your concerns is that our metric measures TD gain rather than e.g. True Gain. It would be extremely surprising for models trained with TD to generalize to other losses. This is visible in Fig 1a where the models trained with MC do not have a nice TD gain curve.
> > > Another concern was that improvements in the metric did not reflect improvements in RL. This is addressed above, but a sub-concern was that it wasn’t clear which behaviour is preferable. This is of course an open question, but an easy target for desirable behaviour that we would like to see for TD methods is the behaviour of SL methods (considering SL methods converge quite quickly to the target V and are known to generalize well), which are identified in Figure 1.
> > >
> > >
> > > Additional worry:
> > > As mentioned in the feedback, the magnitudes of gain, both at 0-offset and elsewhere, are fairly consistent across environments and runs. Also note that the learning rate is kept constant across environments and runs for the figure. We did try dozens of different normalizations, including normalizing by the 0-offset gain, but none proved to be informative.
> > >
> > > (3)
> > > There is a fairly generalized complaint, no pun intended, among RL researchers that current Deep RL methods have extremely poor generalization. A concise statement of the open problem thus seemed unnecessary, but perhaps it bears repeating: “why are DNNs doing decently at generalization in SL, but doing very poorly in RL?”
> > > To rehash the Discussion section: the fundamental problem being raised by the paper is that we *can* measure these differences quantitatively, but we *can’t* explain where they come from. This problem, we believe, requires attention and further research beyond our group.

---

### Official Review · AnonReviewer3 · 2019-10-23
**Official Blind Review #3**

**Rating:** 6

**Review:**

The manuscript is analyzing the "generalization" in TD(lambda) methods. It includes supervised learning from trajectories, on-policy imitation learning, and basic RL setting. Moreover, memoization performance has also been measured. Main conclusion is the fact that TD(0) performs very similar to tabular learning failing to transfer inductive biases between states. There are also additional surprising results about optimization.

The empirical study is rather complete and significant. It raises interesting questions for the community and states some clear open problems.

Results are conclusive and interesting. I believe it is a study which a practitioner using TD-based method should be aware of. Hence, I believe it is impactful.

On the other hand, the manuscript has some significant issues which need to be resolved as follows:

- One major issue is calling the analyzed metric "generalization". Generalization by definition requires something beyond what is seen. I believe the quantity defined in (9) is generalization. However, it can not be computed. Hence, calling its empirical version, "generalization" is confusing and a clear misuse of the term. I strongly urge authors to call the observed quantity something else. "Empirical expected improvement", "gradient regularity", "expected gain", etc. are some candidates come to my mind.

- The optimization aspect is very interesting; however, it confuses the exposition significantly. I think it is better to give all results using adam first, and then showing the comparisons between adam and rmsprop later would be much more readable and easier to understand.

- There are some clarity issues in the explanation of the experiments. Figure 3 is very confusing and it requires multiple reading to be understandable. A clearer visualization or a better explanation would improve the paper.

- I am puzzled about why the authors did not use Q_MC in policy evaluation experiments (Section 3.3). I think it can very well be used in a straightforward manner. It would be an interesting addition to the experiments.

Minor Nitpicks:
- Memorization section is not clear. The discussion on N is very confusing as "14.4% for N = 2 and of 16.1% for N = 50" does not match any of "10.5%, 22.7%, and 34.2%" Can you give full error table in appendix?

Overall, I like the study and suggest to accept it hoping authors can fix the issues I raise during rebuttal period.

**Experience Assessment:**

I have read many papers in this area.

**Review Assessment: Checking Correctness Of Derivations And Theory:**

I carefully checked the derivations and theory.

**Review Assessment: Checking Correctness Of Experiments:**

I carefully checked the experiments.

**Review Assessment: Thoroughness In Paper Reading:**

I read the paper thoroughly.

---

> ### Author Response · Authors · 2019-11-11
> **Feedback**
>
> Dear reviewer, thank you for your feedback. We agree that there are many nuances to our work that make presentation challenging, and any suggestions to improve this are welcome.
>
> > On the metric name
> We acknowledge that calling (9) a measure of generalization conflicts somewhat with existing notions. We gave a lot of thought to what would be an appropriate name for this phenomenon, in the end we settled for the lengthy “gradient update generalization”. “Gradient” ecompasses the parameter sharing aspect of this, “update” refers to the measure being related to a single discrete update, and “generalization” refers to the desire to improve on more than some given samples (which is related to generalization in the classical sense). Would something like “gradient update improvement” or “gradient update gain” make more sense?
>
> > On optimization aspects
> We agree that the exposition of the optimizers adds complexity. The current presentation is meant to be more compact, and to highlight the consistent differences we found between optimizers that are also consistent with the findings of other papers.
>
> > On Figure 3
> We will revise the caption of this Figure as well as the references to it within the text. Here is the current new caption:
> “Figure 3: Policy evaluation on Atari: evolution of the distribution of $\Delta$, the difference since last visit, during training. In (a) the DNN is forced to memorize, as such the density of $\Delta$ is concentrated around 0 (thin red/white band). In (b-c), Q-Learning and Sarsa, the density is much less peaked at 0 (larger yellow/green bands) as the DNN learns about states without visiting them. In (d) the DNN learns quickly presumably without memorizing (the distribution of $\Delta$ is more spread out and not as concentrated around 0, seen by the larger yellow/green band), as it is trained on Monte-Carlo returns, and quickly converges as can be seen by the high density of positive $\Delta$s early. In (e,f) we see the effect of using $\lambda$ returns (see appendix A.6 for all values of $\lambda$).”
>
> > On Q_MC
> Unless Q_MC refers to something else, we already have this experiment. In section 2.2 we consider Q_MC to be a “supervised” task, using eq (10), and present its results in Figure 1a which we simply refer to as MC. Should we replot the MC curve for 1a in Figure 2 for completeness? (both experiments are using the same data)
>
> > Memorization section
> In retrospect the numbers indeed did seem unlikely. We investigated and found a typo in the code. Here is the full table (added in appendix, Table 1, page 13):
>
> D / N     2       10        50
> 10k      5.7      8.7      8.5
> 100k    38.8    45.6    56.5
> 500k    44.7    79.3    85.8
>        (error %)

---

### Official Review · AnonReviewer2 · 2019-10-25
**Official Blind Review #2**

**Rating:** 6

**Review:**

This paper studies the generalization property of DRL.  Fundamentally, this is a very interesting problem. The authors experimentally analyze this issue through the lens of memorization, and showed that it can be observed directly during training. This paper presents the measure of gradient update generalization, best understood as a side-effect of neural networks sharing parameters over the entire input space. This paper is very written, and well organized.  The experiments are quite solid. However  I may be not capable in judging the novelties and contributions of this paper, since I did not conduct research on this topic.


**Experience Assessment:**

I do not know much about this area.

**Review Assessment: Checking Correctness Of Derivations And Theory:**

I did not assess the derivations or theory.

**Review Assessment: Checking Correctness Of Experiments:**

I assessed the sensibility of the experiments.

**Review Assessment: Thoroughness In Paper Reading:**

I made a quick assessment of this paper.

---

> ### Author Response · Authors · 2019-11-11
> **Feedback**
>
> Dear reviewer, we very much agree that this is an interesting problem. If you have any further questions we will be happy to answer them.

---

### Decision · Program_Chairs · 2019-12-19

**Decision:**

Reject

**Comment:**

This paper received three reviews. R1 recommends Weak Reject, and identifies a variety of concerns about the motivation, presentation, clarity and soundness of results, and experimental design (e.g. choice of metrics). In a short review, R2 recommends Weak Accept, but indicates they are not an expert in this area. R3 also recommends Weak Accept, but identifies concerns also centering around clarity and completeness of the paper as well as some specific technical questions. In their response, authors address these issues, and have a constructive back-and-forth conversation with R1, who remains unconvinced about significance of the empirical results and thus the conclusion of the overall paper. After the discussion period, R3 indicated that they weakly favored acceptance but agreed that the paper had significant presentation issues and would not strongly advocate for it. R1 advocated for Reject, given the concerns identified in their reviews and followup comments. Given the split decision, the AC also read the paper. While the work clearly has merit, we agree with R1's comment that it is overall a "potentially interesting idea, but the justification and presentation/quantification of results is not good enough in the submitted paper," and feel the paper really needs a revision and another round of peer review before publication.